# Spatial Analysis of Determinants of COVID-19 Vaccine Hesitancy in Portugal

**DOI:** 10.3390/vaccines12020119

**Published:** 2024-01-24

**Authors:** Constança Pinto de Carvalho, Manuel Ribeiro, Diogo Godinho Simões, Patrícia Pita Ferreira, Leonardo Azevedo, Joana Gonçalves-Sá, Sara Mesquita, Licínio Gonçalves, Pedro Pinto Leite, André Peralta-Santos

**Affiliations:** 1Direção de Serviços de Informação e Análise, Direção-Geral da Saúde, Alameda D. Afonso Henriques, 45, 1049-005 Lisbon, Portugal; constanca.lage@ulsla.min-saude.pt (C.P.d.C.); joao.simoes@arslvt.min-saude.pt (D.G.S.); patriciaferreira@dgs.min-saude.pt (P.P.F.); pedroleite@dgs.min-saude.pt (P.P.L.); 2Unidade de Saúde Pública Alentejo Litoral, Unidade Local de Saúde do Litoral Alentejano, Rua do Hospital Conde Bracial, 7540-166 Santiago do Cacém, Portugal; 3Centro de Recursos Naturais e Ambiente, Instituto Superior Técnico, Universidade de Lisboa, Av. Rovisco Pais 1, 1049-001 Lisbon, Portugal; manuel.ribeiro@tecnico.ulisboa.pt (M.R.); leonardo.azevedo@tecnico.ulisboa.pt (L.A.); 4Unidade de Saúde Pública Almada-Seixal, Agrupamento de Centros de Saúde de Almada-Seixal, Administração Regional de Saúde de Lisboa e Vale do Tejo, Av. Rainha D. Leonor, n° 2, 5°, 2809-010 Almada, Portugal; 5Unidade de Saúde Pública Zé Povinho, Agrupamento de Centros de Saúde do Oeste Norte, Administração Regional de Saúde de Lisboa e Vale do Tejo, Rua Etelvino Santos, 2500-297 Caldas da Rainha, Portugal; 6Social Physics and Complexity Research Group, Laboratory of Instrumentation and Experimental Particle Physics, Av. Prof. Gama Pinto, n.2, Complexo Interdisciplinar, 1649-003 Lisbon, Portugal; joanagsa@lip.pt (J.G.-S.); smesquita@lip.pt (S.M.); 7Nova Medical School, Campo dos Mártires da Pátria 130, 1169-056 Lisbon, Portugal; 8Serviços Partilhados do Ministério da Saúde, Av. Da República 61, 1050-099 Lisbon, Portugal; licinio.goncalves@spms.min-saude.pt; 9Public Health Research Centre, NOVA National School of Public Health, Universidade NOVA de Lisboa, Av. Padre Cruz, 1600-560 Lisbon, Portugal; 10Comprehensive Health Research Centre (CHRC), Universidade NOVA de Lisboa, Rua do Instituto Bacteriológico, n°5, 1150-082 Lisbon, Portugal

**Keywords:** COVID-19, SARS-CoV-2, vaccine hesitancy, social deprivation, migrant health

## Abstract

Vaccine hesitancy tends to exhibit geographical patterns and is often associated with social deprivation and migrant status. We aimed to estimate COVID-19 vaccination hesitancy in a high-vaccination-acceptance country, Portugal, and determine its association with sociodemographic risk factors. We used the Registry of National Health System Users to determine the eligible population and the Vaccination Registry to determine individuals without COVID-19 vaccine doses. Individuals older than five with no COVID-19 vaccine dose administered by 31 March 2022 were considered hesitant. We calculated hesitancy rates by municipality, gender, and age group for all municipalities in mainland Portugal. We used the spatial statistical scan method to identify spatial clusters and the Besag, Yorke, and Mollié (BYM) model to estimate the effect of age, gender, social deprivation, and migrant proportion across all mainland municipalities. The eligible population was 9,852,283, with 1,212,565 (12%) COVID-19 vaccine-hesitant individuals. We found high-hesitancy spatial clusters in the Lisbon metropolitan area and the country’s southwest. Our model showed that municipalities with higher proportions of migrants are associated with an increased relative risk (RR) of vaccine hesitancy (RR = 8.0; CI 95% 4.6; 14.0). Social deprivation and gender were not associated with vaccine hesitancy rates. We found COVID-19 vaccine hesitancy has a heterogeneous distribution across Portugal and has a strong association with the proportion of migrants per municipality.

## 1. Introduction

The COVID-19 pandemic has profoundly impacted societies worldwide, population health, and global economies [1,2,3,4,5,6]. Mass vaccination programs were an essential strategy to tackle the spread of SARS-CoV-2 and reduce severe forms of COVID-19, allowing for total social and economic activity [2,3,5,7,8,9,10]. However, one of the potential threats to this goal was vaccine hesitancy [3,5,9,10,11,12].

Vaccine hesitancy, defined by the Strategic Advisory Group of Experts on Immunization (SAGE) Working Group on Vaccine Hesitancy, means the “delay in acceptance or refusal of vaccination despite the availability of vaccination services. Vaccine hesitancy is complex and context-specific, varying across time, place and vaccines” [13,14]. It encompasses a full spectrum of attitudes and beliefs, ranging from complete acceptance to complete refusal of any vaccine [13,14,15,16,17,18].

The “3 Cs” model (Complacency, Confidence, and Convenience) is commonly used to explain the factors that might influence vaccine hesitancy. Complacency refers to the paradoxical tendency to trivialize the risks of vaccine-preventable diseases precisely because of the success of vaccination campaigns in controlling them. Confidence relates to trust in vaccines, vaccination programs (including health services and professionals), and the authorities and policymakers implementing them. Convenience relates to geographical and physical accessibility, affordability, and the ability to understand and appeal to vaccination campaigns [13,14]. More recently, two new “Cs” have been proposed: “Risk Calculation”, referring to an extensive search for information on vaccination risks and side effects, and “Collective Responsibility”, regarding the importance given to the commitment of individuals to protect their community [19,20].

Another essential tool developed by the SAGE Working Group is the “Vaccine Hesitancy Determinants Matrix”, which classifies vaccine hesitancy determinants into three different groups. Contextual determinants include factors related to social, political, cultural, or economic contexts and environments. Individual and group determinants refer to individuals’ and their social groups’ perceptions of contextual factors. Vaccine- or vaccination-specific determinants deal with factors related directly to the vaccines or vaccination programs [13,14].

During the COVID-19 pandemic, vaccine hesitancy received extra attention. The impact of the pandemic and the characteristics of COVID-19 vaccines—such as their shorter development time and the use of novel antigen carrier methods like mRNA or adenoviruses—may have exacerbated vaccine hesitancy and resistance [1,2,10,11,17].

Portugal, historically, has had an excellent track record of vaccine acceptance and uptake in the population, which translates into a very high vaccination coverage, including in the context of COVID-19 [10,21,22,23,24,25].

However, vulnerable pockets of unvaccinated people might remain since unvaccinated individuals may cluster together [6,10,12,26]. With the plateauing of vaccination uptake rates and the emergence of new variants, it is fundamental to identify, characterize, and monitor clusters of the population that remain hesitant, due to the risk of community transmission remaining active in these groups [3,9,10,12,23,27]. Being a migrant or living in a socially deprived setting has been associated with a lower vaccine uptake because of hesitancy or lack of access [5,9,12,23,26]. This characterization is essential to inform policies and to implement measures aimed at hesitant clusters in order to ensure the immunization of all eligible individuals and achieve lasting control over SARS-CoV-2 transmission [3,5,9,12,23,27,28,29].

Thus, this study aims to estimate the proportion and spatial distribution of COVID-19 vaccine hesitancy in mainland Portuguese municipalities and the effect of age, gender, migrant status, social deprivation, and geographic location.

## 2. Materials and Methods

### 2.1. Data Sources and Study Design

We conducted an ecological study using data from multiple databases aggregated at the municipal level in mainland Portugal. We used the single centralized national electronic vaccination registry (VACINAS) used by all healthcare professionals to determine the number of non-vaccinated individuals among those eligible for vaccination on 31 March 2022 per gender and age group (according to COVID-19 vaccination priority age groups) in each of the Portuguese municipalities [30]. The Ministry of Health coordinated COVID-19 vaccination, and vaccination records were paramount to facilitate non-pharmacological interventions. Hence, this record provides excellent national coverage and representativeness [30,31]. The autonomous island regions of the Azores and Madeira were excluded from this study since they use a different health information system, and the data were not easily accessible.

We used the Portuguese adaptation of the European Deprivation Index (EDI-PT) at the municipality level [32]. Additionally, we used publicly available databases from Statistics Portugal (INE) to identify the proportion of migrants per municipality as of 2020 [33,34,35]. 

### 2.2. Outcome

We defined a vaccine-hesitant individual as an individual eligible for COVID-19 vaccination with no registry of any COVID-19 vaccine by 31 March 2022. All Portuguese residents are eligible for free vaccination, irrespective of nationality [36] and, by this date, all eligible individuals had had the opportunity to be vaccinated [30,37,38]. We excluded deceased individuals and those younger than five years from our eligible population at the end of the study period. 

We evaluated vaccine hesitancy as the proportion of non-vaccinated individuals among eligible individuals registered in the National Registry of National Health System Users (NHS register). Vaccine hesitancy was stratified by gender, age group, and residence municipality.

### 2.3. Portuguese Version of the European Deprivation Index

The explanatory variable for vaccine hesitancy was social deprivation at the municipality level, as measured using the Portuguese version of the European Deprivation Index (EDI-PT). We used the EDI-PT because it is a compound indicator that includes eight different variables related to social deprivation, making it possible to capture the complexity of the concept. Moreover, it has solid theoretical and statistical foundations and is used at the European level. The EDI-PT is available online and is stratified by municipality, parish, and census block group [32,39,40]. We used the EDI-PT score and quintiles as the association between vaccine hesitancy and deprivation was non-linear at visual inspection (Appendix A).

### 2.4. Data on Migrants 

Individuals considered migrant residents met either one of two criteria: (1) individuals with non-Portuguese nationality and legal authorization of residence; (2) individuals with non-Portuguese nationality that had requested legal authorization of residence. Neither of these two groups includes short-term visas, student or work visas, nor those with an irregular authorization status [33,34].

We calculated the proportion of migrants per municipality by summing the individuals resident in each municipality that met either of the two criteria and dividing them by the total population of the respective municipality, as given by the 2020 population estimates from Statistics Portugal.

### 2.5. Statistical Analysis

To characterize the study population and its distribution across the different variables, we conducted a descriptive analysis using measures of central tendency and variability for numeric variables and calculated absolute and relative frequencies for categorical variables. We used age group intervals to match the prioritization criteria for the COVID-19 vaccination campaign [30,36].

We used spatial statistical methods to determine the association between vaccine hesitancy and explanatory variables at the municipality level. To model vaccine hesitancy and its determinants, we used a Bayesian hierarchical Poisson model [41] (i.e., Besag, York, and Mollié model, or BYM) to quantify spatial patterns of vaccination hesitancy risk explained by spatial patterns of area-level explanatory variables (i.e., social deprivation, migration, age group, and gender).

#### 2.5.1. Spatial Clusters

To identify and locate spatial clusters of vaccination hesitancy, we applied the SaTScan method [42]. This method assumes a Poisson model for each municipality’s distribution of vaccine hesitancy. It uses a test statistic based on the log–likelihood ratio to detect the existence of significant spatial clusters. The method employs circles of varying radii centered in municipality centroids to scan and identify areas with significantly higher vaccine hesitancy. The null hypothesis test is that the vaccine hesitancy rate inside the scan area equals the rate outside (i.e., constant risk hypothesis). The alternative hypothesis is that the vaccine hesitancy rate inside the scan area is higher than outside. 

For the test statistic, we estimated *p*-values for log–likelihood ratio tests through 999 Monte Carlo simulations and set a *p*-value < 0.01 to reject the null hypothesis. The most likely cluster area identified was the window with the maximum likelihood. Other clusters with statistically significant log–likelihood ratios were identified as secondary potential clusters. All spatial cluster calculations were performed with the R package SpatialEpi version 1.2.8 [43].

#### 2.5.2. Vaccine Hesitancy Risk Model

We applied the BYM model to map the spatial distribution of relative risks of vaccine hesitancy in Portugal at a municipality level (*n* = 278) [44,45,46]. The spatial model, built under the Bayesian hierarchical model’s framework, incorporated social and demographic covariates and non-spatial and spatial random components. We applied the model with the assumption that the number of vaccine-hesitant individuals observed in municipality i (i=1,2,…,n) follows a conditional Poisson distribution with parameter $E_{i}\theta_{i}$, where $E_{i}$ is the expected count and $\theta_{i}$ is the underlying true relative risk: $$y_{i}|\theta_{i}\sim Po(E_{i}e^{\mu_{i}})$$ with
$$\mu_{i}=z_{i}^{t}\beta +\psi_{i}$$

In the Poisson model, $\mu_{i}$ represents the vaccine hesitancy log relative risk in municipality i, $z_{i}^{t}={\left(1,z_{i1},\ldots ,\;z_{ip}\right)}^{t}$ is a vector of intercept and p social and demographic covariates with corresponding $\beta ={\left(\beta_{0},\;\beta_{1},\;\ldots ,\;\beta_{p}\right)}^{t}$ regression parameters, and $\psi_{i}$ represents a random (effects) component, which can be decomposed into a sum of a spatially structured error term $u_{i}$ and an unstructured random error term $v_{i}$ as follows:$$\psi_{i}=u_{i}+v_{i}$$

For the BYM model, the following priors were specified:$$\beta_{j}∼N(0,\sigma^{2})\;v_{i}∼N(0,\sigma_{v}^{2})\;u_{i}|u_{j,i\neq j}∼N\left(\frac{\sum_{j}w_{ij}u_{j}}{\sum_{j}w_{ij}},\frac{\sigma_{u}^{2}}{\sum_{j}w_{ij}}\right)\;\sigma_{v}^{2}∼logGamma(1,\;0.01)\;\sigma_{u}^{2}∼logGamma(1,\;0.005)$$

The spatial weights $w_{ij}$ are provided by a binary adjacency matrix specifying municipalities i and j are neighbors if they share a common boundary ($w_{ij}=1$) or not ($w_{ij}=0$).

We evaluated the importance of covariates to model vaccination hesitancy risk using posterior mean estimates and 95% credible interval (CI) limits computed from separated BYM models fitted with single covariates and spatial and non-spatial random components. For model selection, additional models with different combinations of covariates were specified and compared by computing the deviance information criteria (DIC). We tested two null models with and without spatial components. Adding the spatially structured random effect, the model captures additional variability (spatially structured), reducing the standard deviation of the estimated mean (∆SD = −0.012), and performs slightly better when compared to the model fitted with just an overdispersion parameter (∆DIC = −0.3). The posterior distributions of all parameters and hyperparameters were obtained using the integrated nested Laplace approximations (INLA) method, R package INLA version 22.12.16 [46,47].

## 3. Results

### 3.1. Description of the Dataset and Study Population

From the original dataset of all individuals registered in the national registry, we removed all those non-eligible for the study (those less than five years old and those whose municipality of residence was in the Azores, Madeira, or was unknown). The final eligible population comprised 9,852,283 individuals (Figure 1), with 1,212,565 unvaccinated individuals, corresponding to 12% of the eligible population.

The distribution of the study population and vaccine hesitancy by EDI-PT quintile, gender, and age group is described in Table 1.

The number of vaccine-hesitant individuals tends to be higher in higher-deprivation quintiles, with a relative frequency of 10.4% in the lowest- and 15.4% in the highest-deprivation quintile. However, the relation between the proportion of unvaccinated individuals and the EDI-PT score does not seem linear (see Appendix A). Female individuals tend to have lower vaccine hesitancy than males, consistent across deprivation quintiles (Appendix A). 

Younger age groups tend to have higher proportions (Appendix A) and higher frequencies of vaccine hesitancy. The difference between the highest-deprivation quintile and the other quintiles is lower in the older age groups (Table 1). This suggests that in age groups with a higher perceived risk associated with COVID-19, social deprivation may play less of a role in vaccination attitudes. Regarding the proportion of migrants per municipality, the mean shows a marked increase in the highest-deprivation quintile, ranging from 3.3 to 4.2% (2.1–4.9% SD) in the first four quintiles and jumping to 11.8% (13.1% SD) in the fifth (Table 1).

In Figure 2 and Figure 3, we show the geographical superposition of the EDI-PT and the proportion of migrants with vaccine hesitancy by municipality. There is a high prevalence of vaccine hesitancy, social deprivation, and migrants in some regions, namely Lisbon and adjacent municipalities and the country’s southwest region. However, we have some areas of high deprivation, low hesitancy, and low migrants, mainly the NUTS (Nomenclature of Territorial Units for Statistics) III region Tâmega e Sousa.

### 3.2. Spatial Analysis

#### 3.2.1. SaTScan Analysis

Using the SaTScan method, we show a primary cluster of the vaccine hesitancy variable around the Lisbon and Tagus Valley region, with secondary clusters in the southern (Algarve) and southwest regions (Alentejo Litoral) of Portugal and a few more secondary clusters scattered in the northern and central regions (Figure 4).

#### 3.2.2. Bayesian Inference Model with Integrated Nested Laplace Approximation (INLA)

We used univariate models, one for each explanatory variable: (1) EDI-PT score; (2) EDI-PT quintiles; (3) gender; (4) age groups; (5) proportion of migrants by municipality. Tables with the outputs of the models are available in the Appendix A. We also specified a multivariate model (6) considering three explanatory variables. Below, we describe the results for all the models (1–6). 

(1)EDI-PT score

The results for this model show that the estimated mean parameter for EDI-PT score is 0.005 (95% CI −0.007 to 0.017), suggesting that there is no statistically significant effect (Appendix A).

(2)EDI-PT quintiles

The results for this model suggest that an increase in social deprivation, as measured by EDI-PT quintiles, contributes to a significant increase in the risk of vaccine hesitancy. In fact, the parameter associated with vaccine hesitancy in municipalities in the fourth quintile (high deprivation) presents an estimated mean of 0.093 (95% CI 0.006 to 0.180) (Appendix A), indicating a significant increased risk of vaccine hesitancy by a factor of 1.0973 $\left(e^{0.093}\right)$ when compared to municipalities in the first quintile (lowest deprivation).

(3)Gender

To model the effect of gender on vaccine hesitancy, we used the proportion of women as the explanatory variable. The estimated mean parameter associated with the proportion of women was 0.024 (95% CI −0.004 to 0.051), which indicates that this variable shows no statistically significant effect (Appendix A).

(4)Age groups

We specified a univariate model for each age group. For the 5–14-year-old group, the estimated mean parameter is 0.036 (95% CI 0.014 to 0.058) (Appendix A), suggesting a significant increased risk of vaccine hesitancy by a factor of 1.0364 $\left(e^{0.036}\right)$. For the 15–19-year-old group, the results suggest no statistically significant effect (estimated mean parameter was 0.036 with 95% CI −0.005 to 0.076) (Appendix A). In the 20–39-year-old group, the estimated mean parameter is 0.023 (95% CI 0.012 to 0.033) (Appendix A), suggesting a statistically significant increase in vaccine hesitancy by a factor of 1.023 $\left(e^{0.023}\right)$. The results are not significant for 40–64-year-olds and 65–79-year-olds (Appendix A, respectively). Finally, for the model fitted with the age group of 80 years or more, the estimated mean parameter for this group is −0.017 (95% CI −0.028 to −0.006) (Appendix A), suggesting a statistically significant reduction in the risk of vaccine hesitancy by a factor of 0.983 $\left(e^{-0.016}\right)$, or a 1.66% (expressed as rate, $100*[e^{-0.016}-1]$) reduction in the risk of hesitancy per 1% increase in the proportion of the population in this age group. 

(5)Proportion of migrants

The results for the univariate model using the proportion of migrants as explanatory variable suggest that an increase in the proportion of migrants contributes to a significant increase in the risk of vaccine hesitancy. The parameter associated with this variable presents an estimated mean of 2.243 (95% CI 1.729 to 2.753) (Appendix A), indicating a significant increased risk of vaccine hesitancy by a factor of 9.42 $\left(e^{2.243}\right)$.

(6)Multivariate model

In the multivariate model, we added age group, EDI-PT quintiles, and the proportion of migrants. We chose the 20–39 age group since it was relevantly impacted by an increased risk of vaccination hesitancy (Appendix A). The covariates of this model were able to explain around 84% of spatial variability in the outcome variable. Table 2 shows an RR of 8.01 (95% CI 4.59 to 14.01) in hesitancy risk for every 1% increase in the proportion of migrants per municipality (Table 2). The specification of the BYM model allowed not only for smoothing, as it captured spatially structured random effects (not considered due to unmeasured risk factors), but also for capturing over-dispersion or extra-variability. Based on this, we consider the use of the BYM model appropriate for this data set. Age group and EDI-PT were not significantly associated with vaccine hesitancy; therefore, their posterior means and 95% CI estimates should not be interpreted.

The spatial representation of the relative risk of vaccine hesitancy in the multivariate model shows again that the southwestern region (Algarve and Alentejo Litoral) is at high risk of vaccine hesitancy (Figure 5). 

The map shown in Figure 6 is similar to the map of clusters calculated using SaTScan analysis above (Figure 4) and shows the probabilities of RR being >1. It has a few advantages over the SaTScan cluster map. Namely, it includes probabilities between 0 and 1 (instead of cluster/non-cluster values for SaTScan). It incorporates the covariates included in the multivariate model (whereas SaTScan only includes age-adjusted vaccine hesitancy standardized incidence ratios (SIRs)). This map clearly shows that, once more, in the Lisbon and Tagus Valley region and the southwest, the probability of an RR > 1 approaches 100%, as well as in a few other municipalities scattered across the territory (Figure 6).

## 4. Discussion

Our study examined a population of 9,852,283 individuals eligible for COVID-19 vaccination in Portugal. Among these individuals, 12% were hesitant to receive the vaccine. We found that vaccine hesitancy was most prevalent in the 5–14 age group, with 39.1% expressing reluctance to be vaccinated. Conversely, the lowest levels of hesitancy were observed among those aged 65–79, with only 5.2% reporting hesitation. Higher-deprivation quintiles tended to have higher vaccine hesitancy rates (10.4% vs. 15.4%). The two methods used to identify clusters (SaTScan and the BYM model) confirmed the existence of hesitancy clusters in the Lisbon area and the southwest regions of Portugal (i.e., Alentejo Litoral and Algarve), and the multivariate model showed an association between the increase in the proportion of migrants per municipality and increased hesitancy, even after adjusting for deprivation, gender, and age (RR 8.01, CI 4.59–14.00). This implies that migrant populations are an essential contributor to vaccine hesitancy rates of a municipality, and this is independent of deprivation level. 

In the context of the European Union (EU), Portugal is one of the countries where the COVID-19 vaccination uptake was the highest, with 86.7% of the overall population having completed a primary vaccination scheme, contrasting with 72.9% in the EU overall, as of 30 September 2023 [48]. 

Compared to previous studies on willingness to be vaccinated in the Portuguese population, this hesitancy followed an important temporal shift: the results of a September 2020 study showed that 25% of Portuguese individuals were unsure or unwilling to be vaccinated against COVID-19 [10] but, by June 2021, only 6.5% refused the vaccine, and only 6.8% were still undecided [27]. These later results align with our overall vaccination hesitancy findings and illustrate the relevance and importance of monitoring vaccine hesitancy before and during a vaccination campaign, since hesitancy may vary over time [5,12].

Several studies and reports have shown lower vaccination rates among migrants and ethnic minorities, both because of higher distrust in health services and authorities or due to inequities in access in terms of language, cultural or socioeconomic barriers, or other problems [5,7,9,17,23,26]. This is especially troubling since COVID-19 disproportionally affects ethnic minorities and vulnerable populations [7,9,26]. A study in the United Kingdom highlighted that patients from ethnic minorities and more deprived areas had higher vaccine decline codes (vaccine hesitancy) [49]. Portugal has a set of policies that make it easy to be a legal resident and hence entitled to all the healthcare benefits provided by the National Health Services (NHS), including vaccination. Moreover, during the COVID-19 vaccination campaign, administrative barriers to vaccination were non-existent for all individuals willing to get vaccinated, including illegal immigrants. As these could be represented in the vaccinated population but not in the assumed eligible population, our study might underestimate vaccination hesitancy among migrant communities. Also, the government implemented, in 2021, some targeted COVID-19 vaccination campaigns for migrant workers but the results of this study suggest that a higher investment in policies explicitly targeted at these communities is fundamental to decreasing vulnerability pockets in the population. Rather than increase stigma over migrant populations, these results should urge policymakers in high-income countries to study and understand the reasons for hesitancy in the migrant population and implement strategies to bridge the hesitancy gap. 

Social deprivation and age showed significant associations with hesitancy in the univariate models. However, the results were not significant in the multivariate model because of the proportion of migrants overshadowing them. Nonetheless, our study suggests that higher social deprivation and younger age groups may also contribute to the risk of hesitancy, even if not as strongly, which is also in agreement with other published studies. In fact, younger age is one of the most frequently cited determinants for increased COVID-19 vaccine hesitancy, probably because of lower risk perception among the youth [1,3,5,10,12,23,26,50]. Moreover, we used a loose inclusion criterion for children: children aged between 5 and 12 were only eligible for vaccination in late December 2021 but this eligibility was limited to children who had not been naturally infected with the virus in the previous six months. Therefore, we likely overestimated the number of eligible children and, consequently, vaccine hesitancy in this age group.

Some research has shown that gender is associated with COVID-19 vaccine hesitancy, with men being more likely to be vaccinated, perhaps because of higher risk perception, similar to older age groups [1,3,5,8,10,11,12,50]. Our results showed a tendency is this direction but no significant differences due to gender.

We also give a comprehensive image of the distribution of vaccine hesitancy across the territory, showing that specific regions have clusters of higher proportions of unvaccinated individuals. The number of vaccine-hesitant individuals and their clustering are worrisome for a high-income European country known for its historically high vaccine uptake and acceptance. This illustrates why vaccine hesitancy should be addressed and taken seriously as a public health concern and that relying solely on countrywide uptake rates might prove insufficient, as was described in the previously published literature [10,12,14,16,17,18,23,26,28,29]. The identification of higher prevalence and risk clusters through spatial analysis methodologies should prove helpful in guiding policymakers [6,12,26].

Our study provides some novel insights. To the best of our knowledge, this is one of the first nationwide studies using whole-country data to model COVID-19 vaccine hesitancy with social deprivation and migrant proportion using a spatial approach. We also used data from a nationwide vaccination registry system that records all administered vaccines in a representative and highly accurate manner. Additionally, we provide novel evidence that the proportion of migrants in a municipality is associated with higher hesitancy; given that Portugal had very few barriers to vaccination, hesitancy drivers could be specific to the individual rather than the context. 

This study has limitations. First, it is an ecological study using aggregated data, which means its conclusions can only be extrapolated to the unit of observation, in this case, municipalities. We were not able to explore individual characteristics that could provide valuable insight (i.e., previous vaccination); however, ecological information still provides relevant information to policymakers. 

Second, including all children 5 years or older is likely to have overestimated hesitancy in this age group but it is unlikely to have affected our conclusions regarding migrant populations. 

Third, we cannot exclude potential biases due to not capturing the presence of illegal migrants, which will likely have underestimated vaccine hesitancy. Also, Portuguese citizens who were not residents and had been vaccinated elsewhere might have been considered hesitant because there was no Portuguese vaccination record in the system, despite the Portuguese Electronic Health Records being regularly checked for non-active users, which we eliminated from our analysis, minimizing this bias.

Fourth, the most recent data on the EDI-PT and migrant numbers were from 2020, while this study refers to a period between 2021 and March 2022. This discrepancy could have introduced some bias; however, during the pandemic, migrant flows decreased due to mobility restrictions and the 2020 estimate should still provide a reliable indicator.

Finally, when defining vaccine hesitancy, we used what is, essentially, vaccine uptake as a proxy. However, according to Dubé et al., these two concepts are different [17,18]. Therefore, this study has limitations in reflecting the full spectrum and magnitude of vaccine hesitancy. However, vaccine uptake is a simple measure that provides valuable information about the population level of protection conferred by the vaccine. 

We think our study results are generalizable to other European countries. We used aggregated data and showed country-specific clusters, yet this study uncovers a significant finding: the proportion of the migrant population is correlated with higher rates of vaccine hesitancy. This key insight is not only relevant within the specific contexts we studied but also has broader implications that can be generalized to inform policymaking. It suggests that policymakers should prioritize strategies to enhance access to vaccinations and reduce barriers—particularly those stemming from belief systems—that contribute to increased hesitancy. This approach is vital to ensuring effective and inclusive public health interventions.

Future research should address the complex connections associated with vaccine hesitancy, being from a socially deprived area, and being a migrant. Especially, what are the most relevant access or social constructs that determine vaccine-seeking behaviors?

## 5. Conclusions

COVID-19 vaccine hesitancy has a heterogeneous distribution across the territory of Portugal, with a cluster in the Lisbon area and the southwest (Algarve and Alentejo Litoral). Moreover, our study shows a strong association between a higher migrant proportion per municipality and higher hesitancy, which was independent of deprivation index and age.

## Figures and Tables

**Figure 1 vaccines-12-00119-f001:**
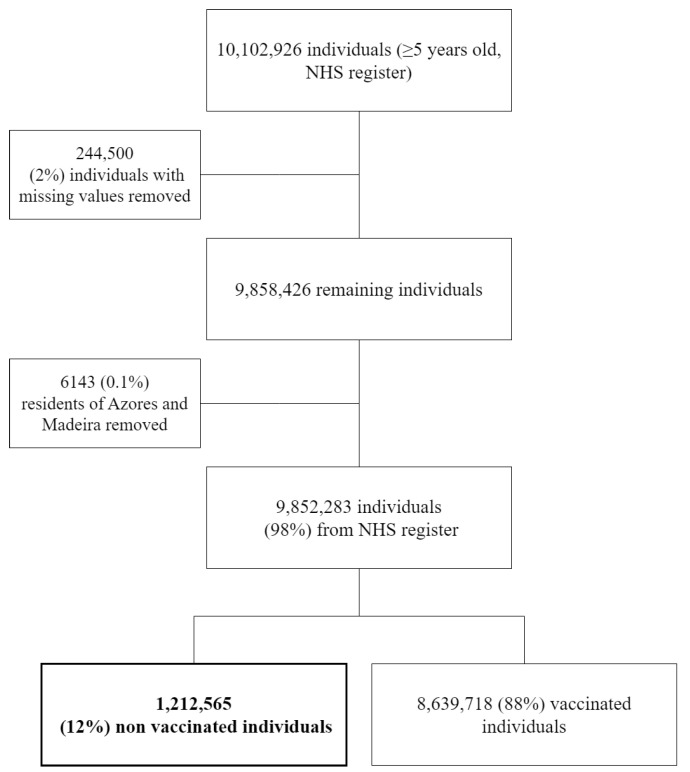
Flowchart of the selection process of the study population.

**Figure 2 vaccines-12-00119-f002:**
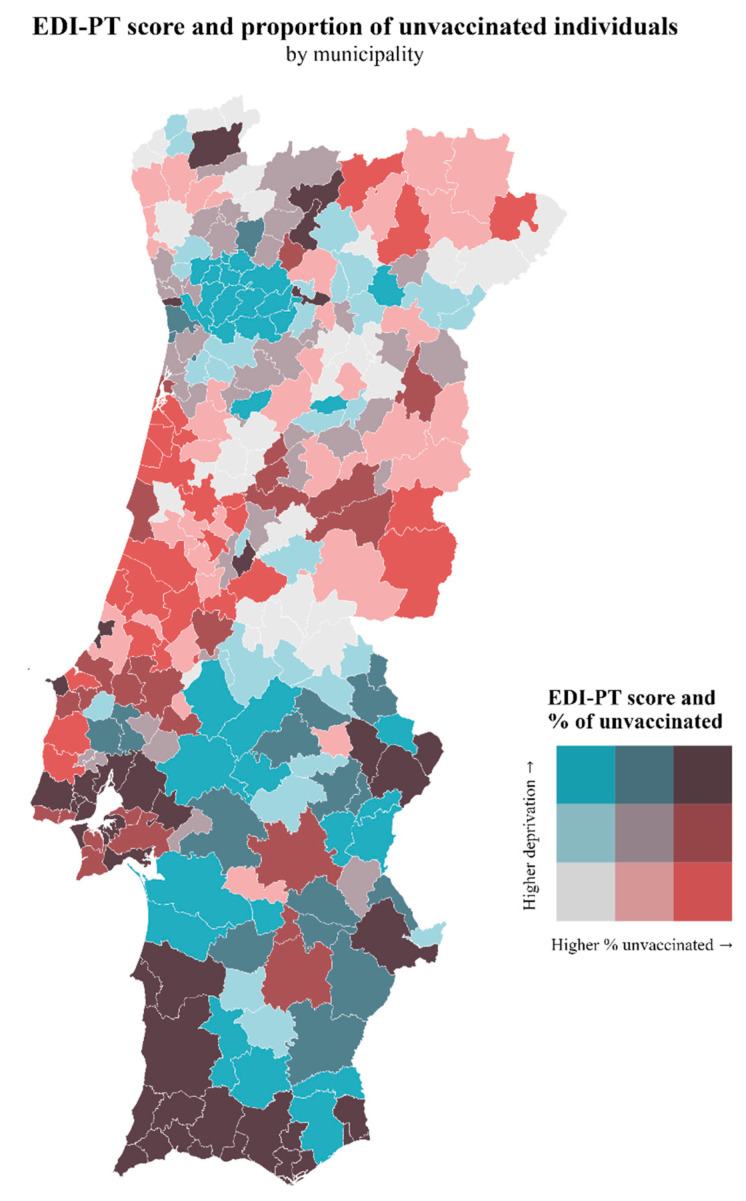
Map of the distribution of the proportion of unvaccinated individuals and EDI-PT score.

**Figure 3 vaccines-12-00119-f003:**
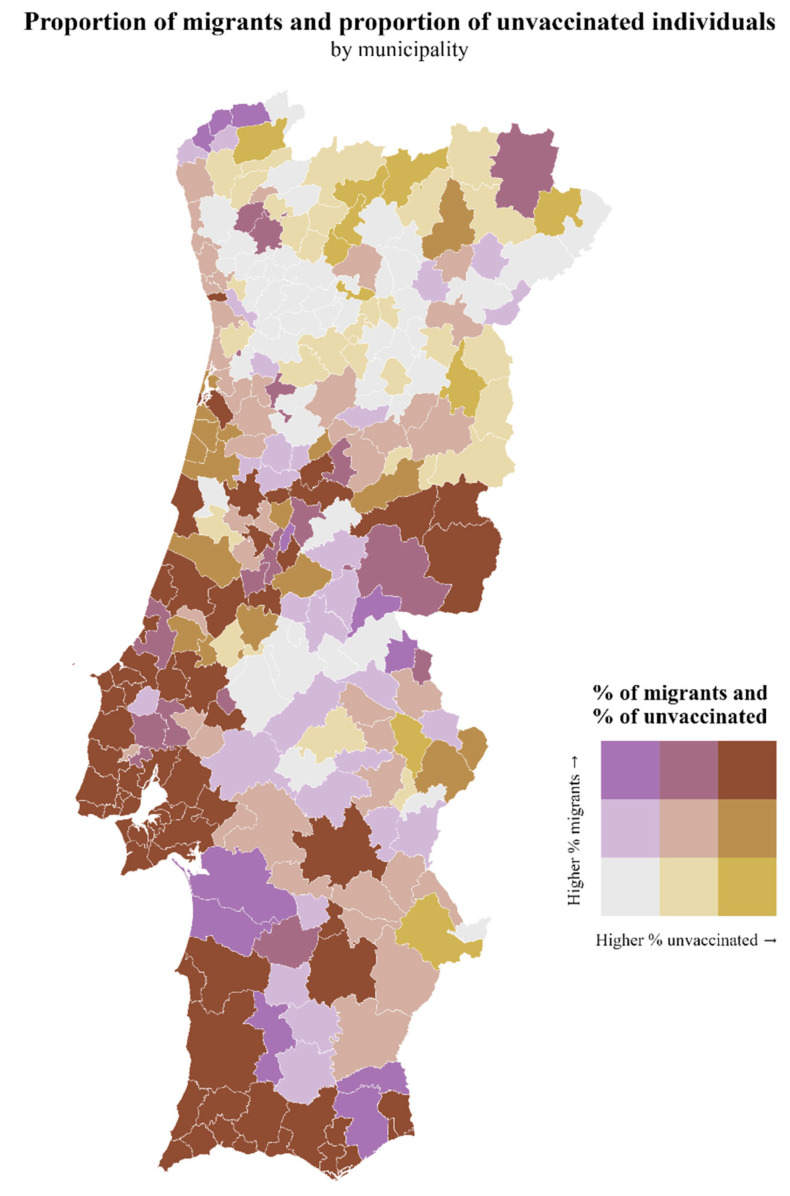
Map of the distribution of the proportion of unvaccinated individuals and the proportion of migrants by municipality.

**Figure 4 vaccines-12-00119-f004:**
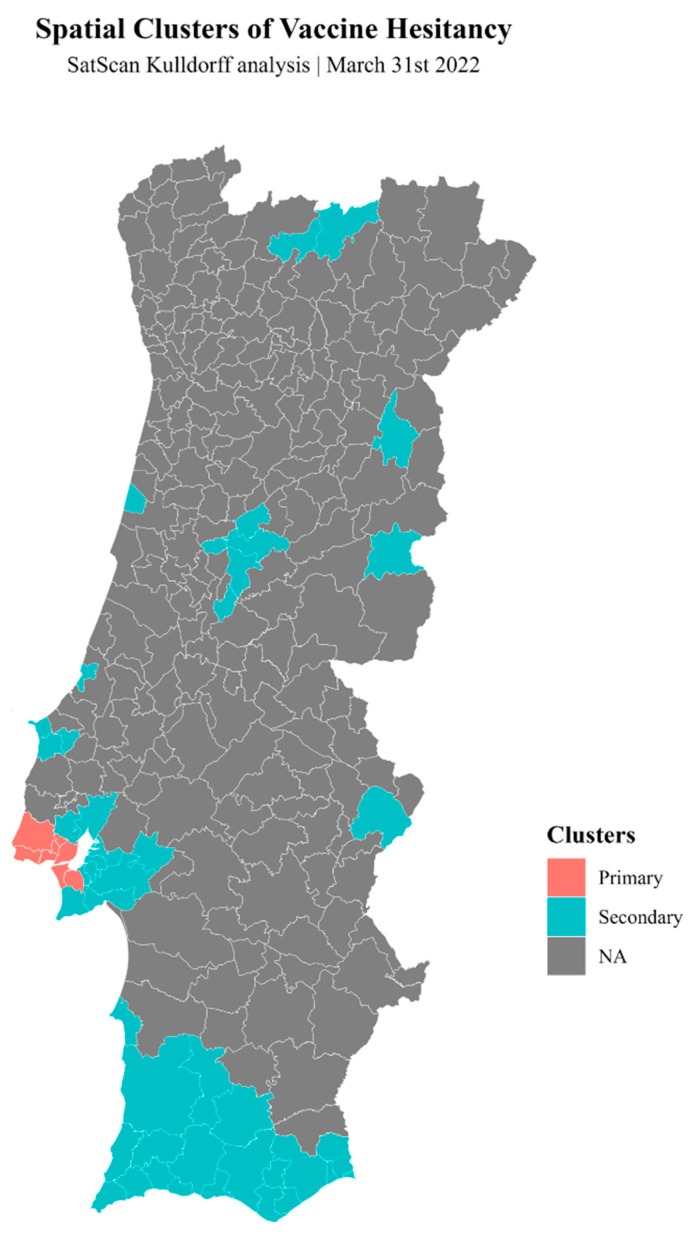
Map showing the spatial clusters of vaccine hesitancy as measured using SaTScan Kulldorff analysis.

**Figure 5 vaccines-12-00119-f005:**
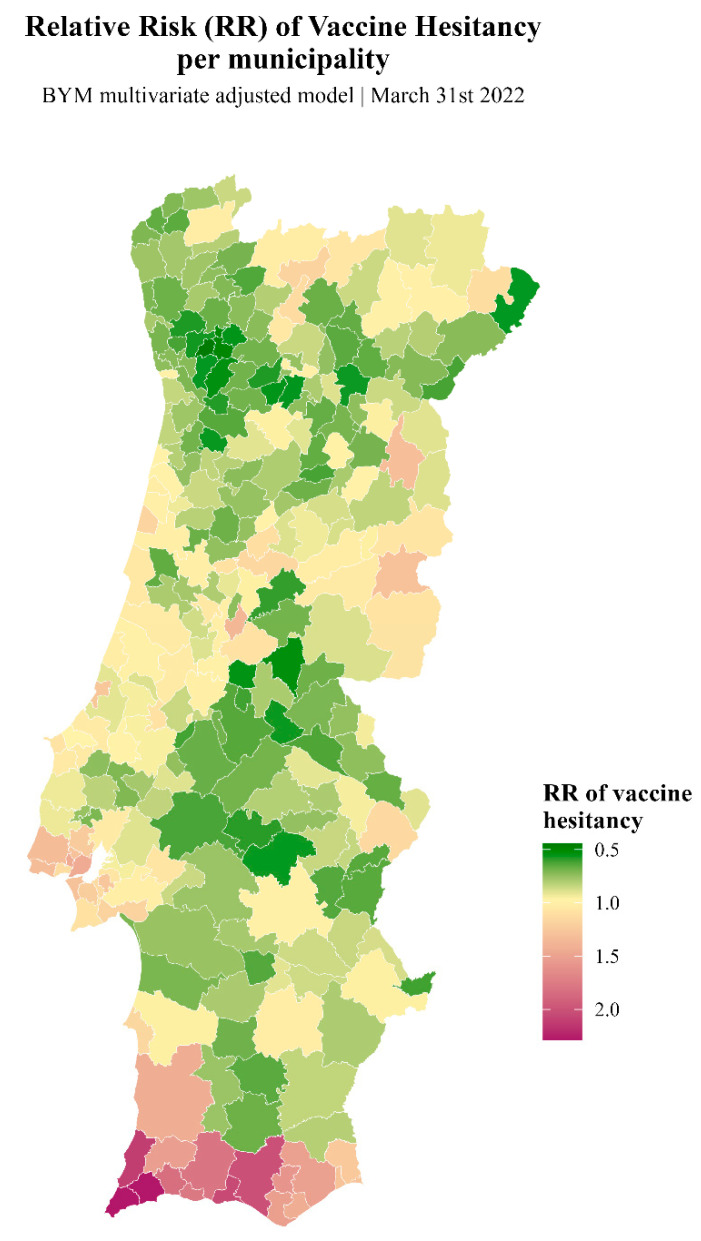
Map showing RR across mainland Portugal, as measured using the adjusted multivariate model.

**Figure 6 vaccines-12-00119-f006:**
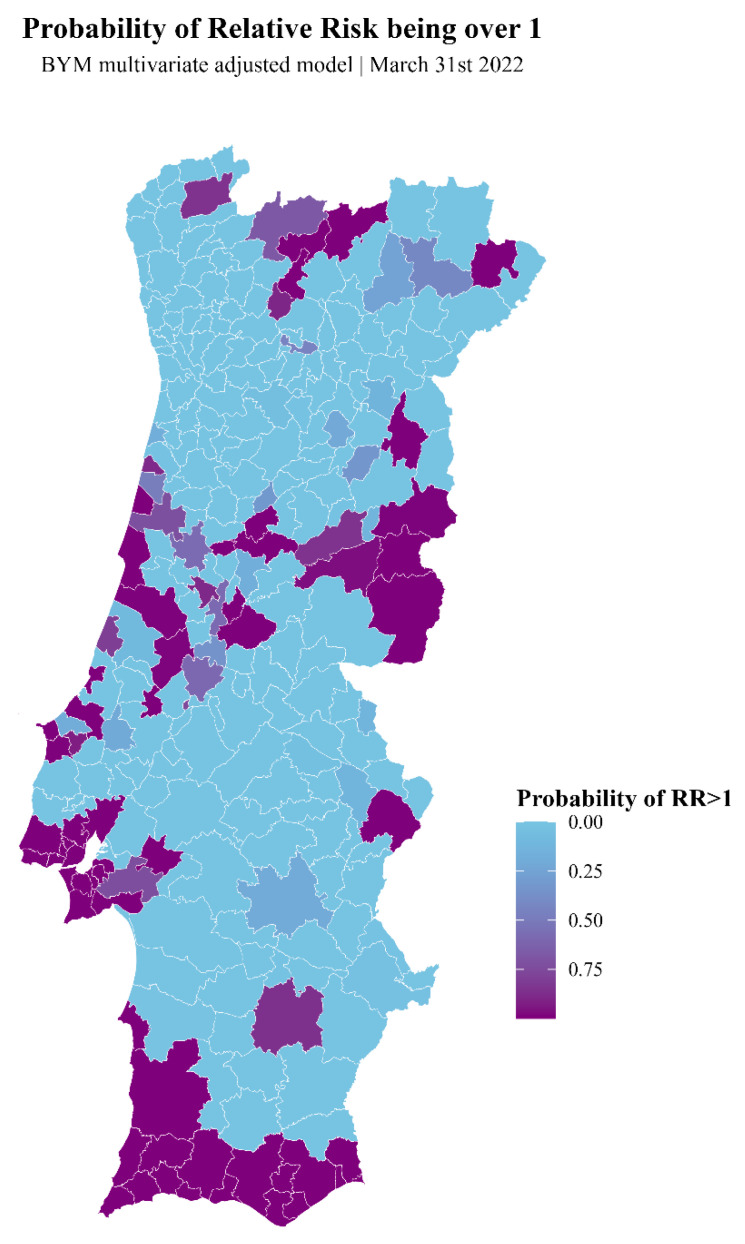
Map showing the probability of RR>1 as measured using the adjusted multivariate model.

**Table 1 vaccines-12-00119-t001:** Characteristics of the study population by deprivation quintile.

Variable	Total	Lowest Deprivation (1st Quintile)	Low Deprivation (2nd Quintile)	Medium Deprivation (3rd Quintile)	High Deprivation (4th Quintile)	Highest Deprivation (5th Quintile)
Eligible individuals	9,852,283	1,363,060	1,450,749	1,415,586	2,877,267	2,745,621
Unvaccinated	1,212,565 (12.3%)	142,041 (10.4%)	156,799 (10.8%)	139,471 (9.9%)	352,285 (12.2%)	421,969 (15.4%)
Gender						
Female	584,862 (11.3%)	67,154 (9.4%)	75,109 (9.8%)	65,730 (8.9%)	170,558 (11.2%)	206,311 (14.2%)
Male	627,703 (13.5%)	74,887 (11.6%)	81,690 (11.9%)	73,741 (10.9%)	181,727 (13.4%)	215,658 (16.6%)
Age group (years)						
5–14	357,402 (39.1%)	39,211 (34.4%)	47,831 (35.9%)	41,085 (32.8%)	104,326 (38.5%)	124,949 (45.9%)
15–19	49,734 (9.7%)	4904 (7.4%)	6156 (8.0%)	5245 (7.2%)	14,453 (9.6%)	18,976 (13.1%)
20–39	348,315 (14.9%)	40,025 (13.2%)	42,507 (12.7%)	36,345 (11.4%)	103,513 (14.8%)	125,925 (18.4%)
40–64	322,613 (8.8%)	39,934 (7.9%)	42,067 (7.7%)	38,978 (7.2%)	93,128 (8.7%)	108,506 (10.8%)
65–79	88,060 (5.2%)	11,371 (4.5%)	11,600 (4.6%)	11,285 (4.5%)	24,010 (5.0%)	29,794 (6.5%)
80+	46,441 (6.5%)	6596 (5.5%)	6638 (6.0%)	6533 (6.1%)	12,855 (6.4%)	13,819 (7.6%)
Prop. of migrants (by municipality)						
Mean % (SD)	5.4% (7.4)	3.3% (2.1)	3.6% (2.6)	3.5% (2.5)	4.2% (4.9)	11.8% (13.1)

Notes: Unvaccinated individuals: eligible individuals without a record of any COVID-19 vaccine by 31 March 2022. Deprivation quintiles: as defined by the EDI-PT, a compound indicator that aims to measure social deprivation. Proportion of migrants: sum of foreign individuals with legal authorization of residence and those that had requested it in each municipality, divided by the respective municipality’s population; does not include short-term, student, or work visas nor those without authorization.

**Table 2 vaccines-12-00119-t002:** Risk of COVID-19 vaccine hesitancy in the multivariate model.

	RR	Low CI	High CI
Intercept	0.67	0.53	0.84
20–39 yrs	1.01	0.99	1.02
EDI-PT quintile 2	1.00	0.94	1.07
EDI-PT quintile 3	1.04	0.97	1.11
EDI-PT quintile 4	1.04	0.96	1.13
EDI-PT quintile 5	1.03	0.94	1.13
Prop. migrants	8.01	4.59	14.00

Notes: RR—relative risk, CI—credible interval, EDI-PT—European Deprivation Index (Portugal).

## Data Availability

The data presented in this study are available on request from the corresponding author.

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
