# Peer review of "Spatial Analysis of Determinants of COVID-19 Vaccine Hesitancy in Portugal"

_vaccines, 2024, doi:10.3390/vaccines12020119_

Round 1
Reviewer 1 Report
Comments and Suggestions for Authors
This is an original paper.
I suggest some minor revisions.
Some relevant literature is missing
- Some systematic reviews:
- DOI: 10.3390/vaccines10081349
- DOI: 10.3390/vaccines9020160
- papers for comparing the spatial analysis
- DOI: 10.1177/00914150221106709
- DOI: 10.4081/gh.2022.1037
- DOI: 10.3390/vaccines10030412
Reviewer 2 Report
Comments and Suggestions for Authors
After careful consideration, I fell that the manuscript entitled “Spatial analysis of determinantes of COVID-19 vaccine hesitancy in Portugal” has merit, but is not suitable for publication as it currently stands. Therefore, my decision is "Major Revision."
Here are my comments:
- line 88. Typo: the reference 8 appears twice
- Line 132 and Figure S1: It's not clear how the proportion of unvaccinated was calculated for each EDI-TP score (on the x-axis). Were these values grouped together? Considering which intervals? Please explain better how this figure was obtained.
- line 152: provide BYM model reference(s)
- Section 2.5.2 (Vaccine hesitancy risk): Describe which prior distribution were adopted in the BYM model. Were the analyses carried out considering informative or non-informative priors?
- line 190 and Figure 1: The text of the manuscript states that individuals <5 years old were removed, while Figure 1 states that individuals >5 years old were initially eligible. Please correct these terms, since it is not clear whether an individual aged 5 is eligible or not.
- Table 1: Is there a specific reason for defining the "15-19" category in age group variable? We can see that it has a much smaller range (and number of observations) than the other groups.
- Table 1, S2 and BYM model: Observing the results in Table 1, wouldn't it be expected that the estimate (mean or 0.5 quantile) of "EDI-PT quintile 4" would be negative, since the percentage of unvaccinated in the 3rd quintile is lower than in the 1st quintile? According to the same rationale, why wasn't the 5th quintile significant (and didn't it have a higher mean than the 3rd quintile), given that the percentage of unvaccinated individuals was higher than the 3rd quintile? Note that the number of eligible individuals in these two quintiles is similar, so it shouldn't be a question of the power of the test. Couldn't this phenomenon be an indication of a poor model fit? The authors could comment on this result and also present a justification in the text to indicate that the use of the BYM model is appropriate for this data set.
- lines 268 to 292: Please check the values described in the text, since they do not match the tables cited. For example, in line 268, the values "9.73 (95CI 0.6% to 19.7%) do not agree with Table S2. Similar divergences were observed in some of Tables S3 to S10.
- Table 2: There is no "p-value" in the Bayesian analysis. Significance can be verified by the presence of "1" in the CI. I suggest removing the p-value column.
- line 324. Typo: I believe the correct is “65-79”
- Figure S3, S4. The terms "masculino" and "feminino" are in Portuguese
Reviewer 3 Report
Comments and Suggestions for Authors
I have conducted a review of your ecological study, and my comment is that it is well-conducted and insightful from several points of view. My only suggestion pertains to the discussion of the identified limitations, as it would be beneficial to expand on how these limitations might impact the internal and external validity of the study. Specifically, you could elaborate on how the use of aggregated data at the municipality level may affect the generalizability of your findings. Additionally, you might discuss in more detail the potential biases arising from not capturing illegal migrants and how this could influence the underestimation of vaccine hesitancy. Furthermore, addressing the possible impact of the discrepancy between the data timeframe and the study period on the study's conclusions could provide valuable insights. Lastly, since you mentioned the distinction between vaccine hesitancy and vaccine uptake, consider discussing the implications of using the latter as a proxy for the former and how it may affect the interpretation of your results. These additional discussions would enrich the understanding of the study's limitations and their potential implications.
Round 2
Reviewer 2 Report
Comments and Suggestions for Authors
I believe that the authors have made the corrections and/or justified the answers satisfactorily and, therefore, I feel that the manuscript is now suitable for publication in Vaccines.